# A push-pull strategy to control the western flower thrips, *Frankliniella occidentalis*, using alarm and aggregation pheromones

**Chul-Young Kim, Falguni Khan, Yonggyun Kim** *

Department of Plant Medicals, College of Life Sciences, Andong National University, Andong, Republic of Korea

* hosanna@anu.ac.kr

## Abstract

Since the first report in 1993 in Korea, the western flower thrips, *Frankliniella occidentalis*, has been found in various crops throughout the country. Although more than 20 different chemical insecticides are registered to control this insect pest, its outbreaks seriously damage crop yields, especially in greenhouses. This study developed a non-chemical technique to control *F. occidentalis* infesting hot peppers cultivated in greenhouses. The method was based on behavioral control using an alarm pheromone ("Push") to prevent the entry of the thrips into greenhouses and an aggregation pheromone ("Pull") for mass trapping inside the greenhouses. The greenhouse fences were treated with a wax formulation of the alarm pheromone and a yellow CAN trap covered with sticky material containing the aggregation pheromone was constructed and deployed inside the greenhouses. Field assay demonstrated the efficacy of the push-pull tactics by reducing thrips density in flowers of the hot peppers as well as in the monitoring traps. Especially, the enhanced mass trapping to the CAN trap compared to the conventional yellow sticky trap led to significant reduction in the thrips population. This novel push-pull technique would be applicable to effectively control *F. occidentalis* in field conditions.

## 1. Introduction

The western flower thrips, *Frankliniella occidentalis* (Pergande) (Thysanoptera: Thripidae), is one of the most devastating insect pests to many horticultural crops including more than 500 species spanning 50 plant families, especially those cultured in greenhouses [1–3]. Both the larval and adult stages cause damage to plants by directly feeding on leaves or flowers [4, 5]. Especially, adults transmit plant viruses including tomato spotted wilt virus (TSWV) [6, 7]. TSWV infections have become serious in several horticultural products in Korea, including the high-value hot pepper crop [8]. This pest, originally native to North America, has spread to more than 60 countries since the late 1970s, including Europe, Asia, and Australia [9]. It was first observed on Jeju Island in 1993 and now has been spread to most regions in Korea [10].

Different reproductive modes along with a short lifecycle allow the thrips population to rapidly increase during early and late seasons, in which female progeny develop from fertilized

**Funding:** This work was conducted with the support of the Cooperative Research Program for Agriculture Science & Technology Development (Project No. PJ01578901) funded by the Rural Development Administration, Republic of Korea. The funders had no role in study design, data collection and analysis, decision to publish, or preparation of the manuscript.

**Competing interests:** The authors have declared that no competing interests exist.

eggs while males develop from unfertilized eggs [11, 12]. The nutrient-based oogenesis of *F. occidentalis* is well-programmed in females by gonadotropic hormones including juvenile hormone, ecdysone, and prostaglandin [13, 14]. To suppress thrips outbreaks, more than 20 kinds of chemical insecticides including organophosphate, carbamate, pyrethroids, neonicotinoids, avermectin, and spinosad are registered and applied to different agricultural crops [15]. However, the control efficacy against *F. occidentalis* is often unsatisfactory due to its cryptic ability to avoid exposure to the chemical spray, and the development of insecticide resistance [16]. A resistant population to emamectin benzoate showed a 356-fold increased resistance by a specific mutation in the target site and an over-expression of a detoxifying enzyme compared to a susceptible strain in Korea [17]. As alternative control tactics, the use of natural enemies, microbial pathogens, and other biological control agents has been proposed under an integrated insect pest management (IPM) program against *F. occidentalis* [18]. In this IPM program, behavioral control using pheromones has been considered in the monitoring and mass trapping of thrips [19].

Chemical communication using pheromones plays a crucial role in mediating *F. occidentalis* populations [9]. A contact pheromone, 7-methyltricosane produced by male adults, acts as a territorial signal by eliciting a fight signal to other males or gives the location of a male to females [20]. Kirk and Hamilton [21] suggested the presence of male-producing sex pheromones attracting only females. The aggregation pheromone of *F. occidentalis* was identified as containing two components, neryl methylbutanoate (NMB) and lavandulyl acetate (LA) [22]. Kirk et al. [23] suggested another type of chemical communication in mating, an anti-aphrodisiac pheromone produced by males during mating to prevent the multiple mating of females [24]. *F. occidentalis* larvae release an alarm pheromone in an anal droplet containing decyl acetate (10:Ac) and dodecyl acetate (12:Ac), in which the two components are present in a relative mass ratio of 0.4:1 to 1.1:1, respectively [25]. This alarm signal promotes the vigilance of thrips as well as takeoff or refuge-seeking behaviors to survive predator attacks [26]. Moreover, the relative ratio of the two components and their total amounts vary with the hazard intensity by increasing the relative amount of 12:Ac or the total alarm pheromone concentration in the droplet, suggesting that the alarm signal is context-dependent [27].

Using aggregation and alarm pheromones, this study devised a push-pull control strategy against *F. occidentalis* in greenhouse conditions. Fence treatment with alarm pheromone was used to prevent the entry of thrips into the greenhouse while an aggregation pheromone was used for mass trapping inside the greenhouse. To investigate this control strategy, we tested the active pheromone compounds and determined their blend ratio in the laboratory. For field assays, both pheromones were formulated for sustainability in field conditions. Finally, the push-pull tactic was applied to a specific farm without chemical sprays and the control efficacy was compared to that of farms that followed a conventional insecticide spraying program for thrips.

## 2. Materials and methods

### 2.1. Thrips rearing

A laboratory colony of *F. occidentalis* that originated from a local population in Jeonju in Korea was donated by the Rural Development Administration (Jeonju, Korea). It was maintained under 25 ± 1˚C constant temperature and a 16:8 h (L:D) photoperiod with a relative humidity of 60 ± 5%. Another laboratory colony of *F. intonsa* was obtained from a field collection in a hot pepper field in Geosan (N36.81 E127.78, Korea) and reared in the laboratory conditions. All thrips were fed a kidney bean (*Phaseolus coccineus* L.) diet according to the method described in our earlier study [12].

## 2.2. Pheromones

Lavandulyl acetate (LA), lavandulyl methylbutanoate (LMB), and neryl methylbutanoate (NMB) were synthesized (AD, Inc., Andong, Korea) (S1 Fig). Briefly, LA and LMB were synthesized from the precursor, lavandulol ($\geq$ 90%, Sigma-Aldrich Korea, Seoul, Korea) in dichloromethane, which was reacted with acetyl chloride to yield LA, and reacted with isovaleryl chloride ($\geq$ 98%, Sigma-Aldrich Korea, Seoul, Korea) to yield LMB, respectively, under the catalytic activity of trimethylamine. NMB was used in a reaction with geraniol ($>$ 98%, Sigma-Aldrich Korea) and isovaleryl chloride under the catalytic activity of trimethylamine. All products were extracted with ethyl acetate and produced 70 ~ 75% yields. The purity of the compounds was analyzed by gas chromatography (8860 GC, Agilent, Santa Clara, CA, USA) with a DB-1 column (15 m × 0.350 mm, Agilent), an oven temperature of 280°C, and a flow rate of 1.0 mL/min, and was determined to be 98.1% for LA, 98.1% for LMB, and 99.3% for NMB. All synthesized products were in the form of a racemic mixture. Two alarm pheromone components (decyl acetate and dodecyl acetate, 98%) were purchased from Sigma-Aldrich Korea and dissolved in hexane to prepare different concentrations.

## 2.3. Construction of CAN trap

The CAN trap consisted of a short supporting bar (1 m), an empty can (diameter 17 cm and height 18.5 cm), and a sticky plastic bag (see Fig 2). The short supporting bar and the empty can were connected with nails and attached to another long supporting bar (1 m) at an appropriate position for the trapping height. To mix the pheromone with sticky material, 2 mg of pheromone components and 12 g of sticky material (Tanglefoot, Contech Electronics, Rochester, NY, USA) were dissolved in 6 mL of hexane. This mixture was then poured to a sticky plastic zipper bag (30 × 30 cm, HG Pack, Gunpo, Korea). After spreading the sticky mixture with a roller, the bag was used to cover the can by inside-out inversion.

## 2.4. Wax formulation of alarm pheromone

The alarm pheromone (a mixture of decyl acetate and dodecyl acetate in 1.5:1, g/g) was formulated in wax by the method of Kim et al. [28]. Briefly, paraffin wax (Merck, Rahway, NJ, USA) was dissolved and mixed with an emulsifier (Almax 3600, Illshinwells, Seoul, Korea), α-tocopherol (96%), and jojoba oil (see Fig 5A). After the alarm pheromone was added, the slurry was cooled to solidify.

## 2.5. Aggregation behavior test using Y-tube olfactometer

A Y-type olfactometer (the main Y-tube length, 5 cm; two branches 2 cm long; a 45° angle between the branches; inner branch diameter, 5 cm) was placed in a dark room at 25 ± 1°C to avoid visual cue for the choice test. A flow of clean (charcoal-filtered) air at a rate of 0.6 L/min was split and passed through two glass vessels containing either an odor source or control (hexane) stimuli and entered each of the branches of the Y-tube. An air supply system (Power Air Pump, Seoul, Korea) was used for air filtration and flow rate control. Before and after the bioassay, the Y-tube was cleaned with 100% methanol. Test thrips were used three days after adult emergence. Each test used 10 ~ 30 individuals of each sex and was replicated four times by changing the source and control for replication. The response was positive when the thrips passed through the midpoint of the branches for 10 min. Otherwise, the thrips' response was recorded as "no response."

## 2.6. Alarm pheromone test using dispersal behavior assay

An arena was designed using a Petri dish (10 × 3 cm, SPL Life Science, Seoul, Korea) to test the dispersion behavior of the thrips. Test thrips (20 individuals per replication) on the bean diet were put into the center of the dish. A disc (3-mm diameter filter paper) was put near the diet and 10 μL of test solution in hexane was added. After 1 min, the thrips dispersed away from the bean were counted. Hexane was used as the control. Each treatment was replicated three times.

## 2.7. Field test for alarm pheromone

A greenhouse (220 m$^2$) cultivating hot peppers was used in this test. *F. occidentalis* was heavily infesting the peppers. In the middle of the greenhouse, test hot peppers in pots (one plant per pot) without any thrips were randomly assigned by a randomized block design with three replications. Each test plant was covered with a plastic cage (45 × 45 × 45 cm), which had large openings in five directions to allow the entry of nearby thrips. The wax-formulated pheromone (5 g) was loaded in a small pheromone cage (Green Agrotech, Kyungsan, Korea), which was hung at the opening gates of the cage. An empty pheromone cage was used as the control.

The duration of alarm pheromone efficacy was tested by harvesting the lure in the field conditions once a week. The efficacy test was conducted using the dispersion behavior assay described above.

## 2.8. Field test for push-pull tactics

A test greenhouse (660 m$^2$) cultivating about 200 hot peppers was located in Gudam (GD, N36.54 E128.46), Andong, Korea. Alarm pheromones in the wax formation were applied by pasting each (5 g) to the fence around the greenhouse at 2-m intervals for pushing the thrips immigrating from outside. The alarm pheromones were re-applied every four weeks. Twenty CAN traps containing aggregation pheromone were installed inside the greenhouse for pulling the thrips. The sticky bags were replaced every week. During the assay, no chemical insecticides were applied. For reference, thrips presence was compared in two other fields cultivating hot peppers. One was a greenhouse in Hahoe (HH, about 4 km from GD) and an open field in Songchun (SC, 16 km from GD) (Fig 6A). Both fields were frequently treated with chemical insecticides including spinosad during the assay.

## 2.9. Thrips monitoring test in the field

Three yellow sticky traps (Green Agrotech) were installed inside the greenhouse for monitoring changes in the thrips population from March to July. Thrips were identified based on the morphological characters described in an earlier study [12]. Number of *F. occidentalis* in flowers was counted from 10 randomly selected flowers with three replications in each field. The monitoring was performed once a week for one month (June 13 ~ July 13). TSWV infection rates in the hot peppers were measured by ring-spot disease symptoms on leaves and fruits. Each experimental unit was 100 randomly chosen hosts and was replicated three times. To determine TSWV-infected thrips, multiplex polymerase chain reaction (PCR) analysis (see below) was used to discriminate viruliferous thrips. In each site/collection, 30 thrips were used to analyze the viruliferous thrips for the period.

## 2.10. Molecular diagnosis of TSWV using multiplex PCR

A single insect was homogenized in 35 μL of an RNA extraction solution (LGC Bioresearch Technologies, Hoddesdon, UK) in a 1.5-ml sample tube. The homogenate was incubated at

95˚C for 5 min and centrifuged at 13,500 × *g* for 5 min to obtain supernatant. To extract RNA from hot peppers, a piece of a leaf (1 g) was homogenized in 250 μL of a lysis buffer in the Viral DNA/RNA Extraction Kit (Intron Biotechnology, Sungnam, Korea) and RNA was extracted according to the manufacturer's instruction. PCR amplification was performed in a total volume of 20 μL containing 4 μL of template RNA, 8 μL of primers, and 8 μL of SuPrime Script RT-PCR Premix (2X) (Genetbio, Daejeon, Korea). Seven primers [12] were designed for the TSWV (5′-TGTCTAAGGTTAAGCTCACTAAGGAA-3′ and 5′-TTAAG-CAAGTTCTGCAAGTATTGCCTG-3′), *F. occidentalis* (forward primer, 5′-GGT CGC TTC ACC GCT TCC CG-3′), *F. intonsa* (forward primer, 5′-GAC CAG ACT GTT CCG AG A-3′), a common reverse primer (5′-CTC TCC TGA ACW GAG GCT G-3′), and *T. tabaci* (5′-TCT AAA CAG AGG GAA AGG TG-3′ and 5′-AGT GTG CCA ACA AGG CAA TG-3′). Reverse transcription was performed at 50˚C for 30 min. After a denaturation step at 95˚C for 5 min, the subsequent PCR was proceeded according to the following temperature cycle program: 35 cycles of 95˚C for 30 s, 55˚C for 30 s, and 72˚C for 1 min. The PCR products were electrophoresed on 2% (w/v) agarose gel containing 1 × TAE buffer and were visualized by UV light on a Gel Doc imaging system (Bio-Rad, Hercules, CA, USA). The resulting PCR product sizes were 777 bp for TSWV, 287 bp for *F. occidentalis*, 367 bp for *F. intonsa*, and 417 bp for *T. tabaci* [12].

### 2.11. Statistical analysis

Percentage data were arcsine-transformed and confirmed in normality using PROC UNIVARIATE in the SAS program [29]. Data obtained from the Y-tube or choice test were subjected to a one-way analysis of variance (ANOVA) using PROC GLM. Field assay data were assessed by two-way ANOVA. The means were compared using the least significant difference (LSD) test or Student's t test at Type I error of 0.05. Frequency data were analyzed by Chi square test using PROC FREQ.

## 3. Results

### 3.1. Optimal pheromone composition to attract thrips

The ability of two components (LA and NMB) of the aggregation pheromone to attract thrips was tested using two thrips, *F. occidentalis* and *F. intonsa* (Fig 1). In the behavioral assays, methylbutanoate lavandulol (LMB) was used as the control (Fig 1A). LA and NMB were significantly ($F = 73.50$; df = 1, 6; $P = 0.0001$ for LA; $F = 19.11$; df = 1, 6; $P = 0.0047$ for NMB) effective in attracting adult female *F. occidentalis* and male *F. intonsa*. However, LMB did not significantly attract thrips. Then, we prepared blends containing different ratios (1:2, 1:4, and 1:7, g/g) of LA and NMB and compared their attractiveness (Fig 1B). Except for the 1:2 ratio blend, the other two blends effectively attracted both thrips species. Even though males did not show significant preferences, *F. occidentalis* females were significantly ($F = 96.43$; df = 1, 6; $P < 0.0001$) attracted to the 1:7 blend whereas *F. intonsa* females were significantly ($F = 27.77$; df = 1, 6; $P = 0.0019$) attracted to the 1:4 blend (Fig 1).

### 3.2. Development of the yellow CAN trap containing aggregation pheromone

A yellow sticky trap (PLATE-type) has been used to monitor thrips. To enhance the trapping efficiency, a CAN-type trap covering an empty can with a sticky zipper bag (CAN-type) was designed to attract thrips from all directions because the PLATE-type attracted only to the front and back sides. When the attractiveness of the PLATE-type trap was compared to that of

**(A)**

| Components | Response[1] of *F. occidentalis* | | Response[1] of *F. intonsa* | |
|---|---|---|---|---|
| | Females | Males | Females | Males |
| **NMB** | + | NS | NS | + |
| **LA** | + | NS | NS | NS |
| **LMB** | NS | NS | NS | NS |

| LA:NMB (g/g) | Response[1] of *F. occidentalis* | | Response[1] of *F. intonsa* | |
|---|---|---|---|---|
| | Females | Males | Females | Males |
| **1:2** | NS | NS | NS | NS |
| **1:4** | + | + | + | + |
| **1:7** | + | + | + | + |

[1]+represents significant ( $P < 0.05$) preference for the test compound and NS indicates no significant difference.

**(B)**

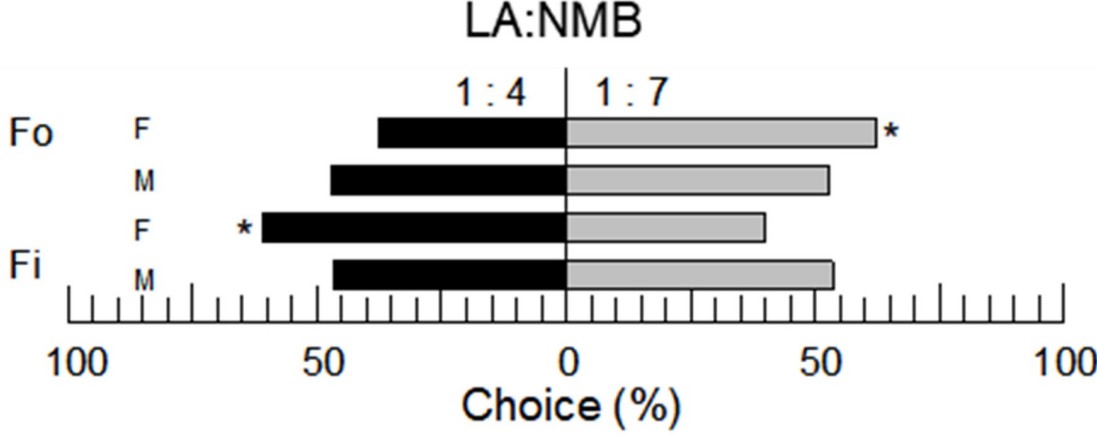

**Fig 1. Differential preference analysis of two flower thrips (*F. occidentalis* and *F. intonsa*) for aggregation pheromone using a Y-tube olfactometer.** (A) Screening of active components and mixture ratios in the aggregation pheromone. The pheromone components were identified as LA (lavandulyl acetate), LMB (lavandulyl methylbutanoate), and NMB (neryl methylbutanoate). (B) Choice tests of the thrips against two pheromone blends. Each response (experimental unit) used 20 adults (< 3-days-old after emergence). Each treatment was replicated four times. The detailed data are presented in S1 Fig.

the CAN-type trap without any aggregation pheromone treatment, they were not significantly different based on the number of thrips per sticky unit area (Fig 2A). However, the trap color significantly ($F = 5.63$; df = 4, 10; $P = 0.0123$) influenced thrips trapping, in which attractiveness was nearly lost in black traps (Fig 2B). The addition of the aggregation pheromone to the CAN trap significantly ($F = 11.34$; df = 1, 4; $P = 0.0028$) enhanced attractiveness even in a

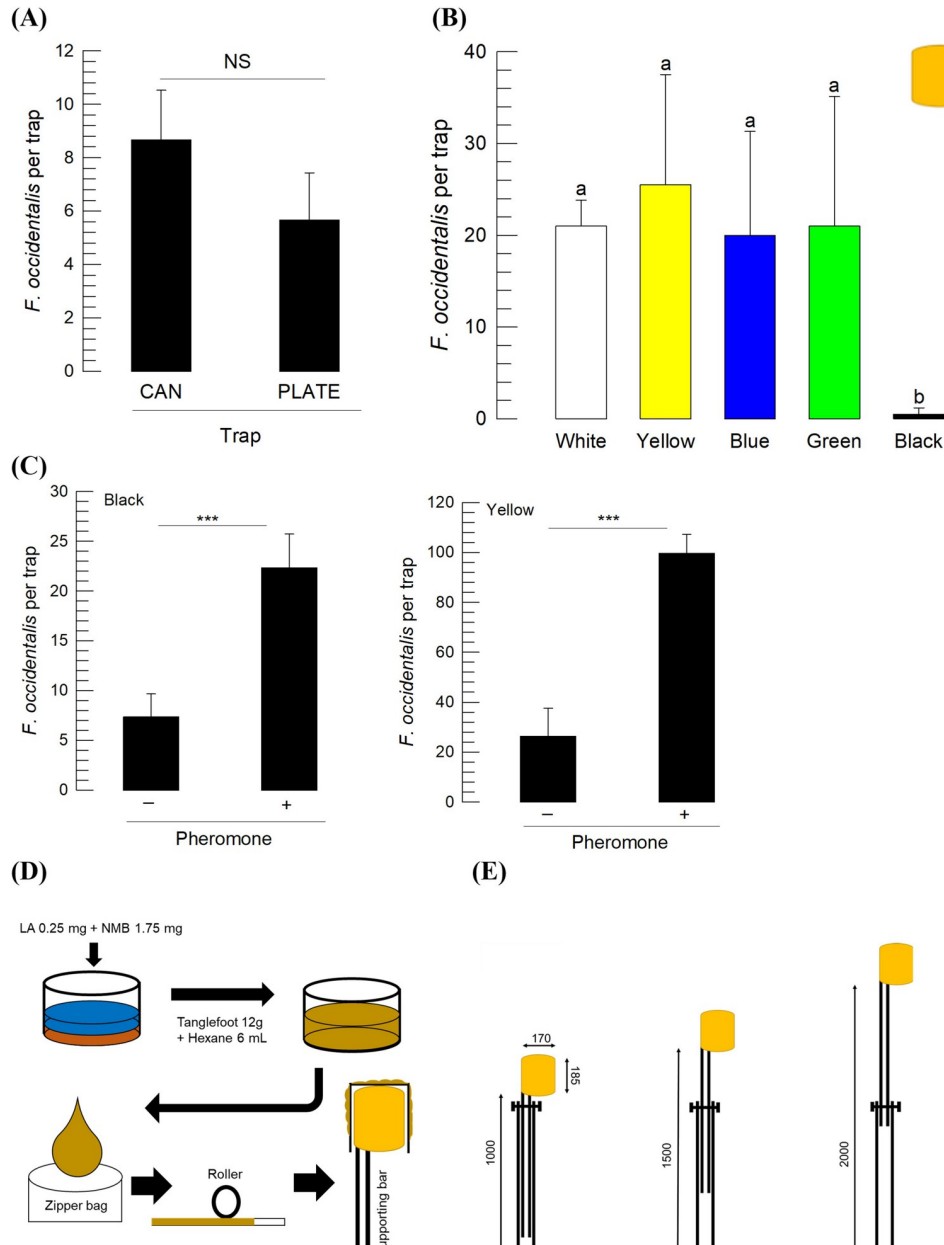

**Fig 2. Development of a yellow CAN trap (CAN) to attract *F. occidentalis* using the aggregation pheromone.** (A) Comparison of the attractiveness of the CAN trap without aggregation pheromone and a yellow sticky trap (PLATE). (B) Effect of CAN trap color on attracting thrips. (C) Effect of aggregation pheromone on attracting thrips using the CAN trap, in which the black CAN trap demonstrated a pheromone effect while the yellow CAN trap demonstrated both pheromone and color effects. Pheromone components were prepared by mixing lavandulyl acetate (LA) and neryl methylbutanoate (NMB) at a 1:7 mass ratio. Each experimental unit represented a trap and was replicated three times. (D) Preparation of a sticky bag for the CAN trap. (E) Dimensions (mm) of the CAN trap parts and adjustment of the CAN trap height.

black background (Fig 2C). With a yellow background, the addition of the aggregation pheromone increased the attractiveness of the CAN trap by almost five-fold compared to the trap without the pheromone. Based on the findings, a yellow CAN trap was constructed (Fig 2D).

For the CAN trap, LA (0.25 mg) and NMB (1.75 mg) were mixed with a sticky material (12 g of sticky material in 6 mL of hexane). The mixture was then poured into a zipper bag and spread using a roller. When the sticky bag was applied to the CAN trap, the sticky inside was inverted to the outside. The height of the CAN trap was controlled by the length of the supporting bar (Fig 2E).

### 3.3. Trapping efficiency of CAN trap in field conditions

The trapping efficiency of the CAN trap for *F. occidentalis* in field conditions was compared to that of a yellow sticky plate (PLATE), which is currently used to monitor thrips including *F. occidentalis* (Fig 3). When we compared the trapping efficiency of these two traps for a week (June 8 ~ June 15), the CAN trap showed > 4-times higher efficiency than the PLATE-type (Fig 3A). Then, we analyzed the number of *F. occidentalis* collected by the two traps over five successive weeks (Fig 3B). The collection numbers were normalized by sticky unit area because of their different sizes. Except for one week in which thrips occurred at a low density, the CAN trap was much more attractive than the PLATE trap ($t$ = 5.29; df = 24; $P$ < 0.0001).

Unexpectedly, the CAN trap was also highly efficient ($t$ = 9.99; df = 24; $P$ < 0.0001) in attracting the whiteflies of *B. tabaci* compared to the PLATE-type. Thrips collected by the CAN trap were classified into different species and the composition was compared to that of the thrips collected by the PLATE trap (Fig 3C). The CAN trap containing the *F. occidentalis* aggregation pheromone attracted all three thrips species of *F. occidentalis*, *F. intonsa*, and *T. tabaci* but it attracted more *F. occidentalis* than the PLATE trap without pheromone ($X^2$ = 14.03; df = 2; $P$ = 0.0009).

### 3.4. Development of thrips repellent using alarm pheromone

In response to the alarm pheromone (10:Ac and 12:Ac = 1: 1.5, g/g), larvae and adults exhibited dispersal behavior within a few seconds (Fig 4A). This dispersal behavior was also observed in response to individual components, in which 12:Ac induced the behavior in both stages, whereas 10:Ac was effective only in adults (Fig 4B). Dispersal behavior to the alarm pheromone components was highly induced in adults compared to larvae. The two-component blend was more effective in inducing the dispersal behavior than single component treatments except for the comparison of 12:Ac and the blend in the adult stage. To clarify the effectiveness between the single-component (12:Ac) and the two-component blend, different doses were applied and the efficacy of inducing dispersal behavior was assessed (Fig 4C). The two-component blend was much more effective at low doses compared to the single component in both larvae and adults. In addition, the two-component blend was effective in inducing the dispersal behavior of *F. intonsa* (Fig 4D). However, the dispersal behavior was highly induced in *F. occidentalis* compared to *F. intonsa* at low doses. These results indicate that the alarm pheromone induced the dispersal behavior of thrips in both larvae and adults and acted as a repellent because the thrips fled the pheromone source.

The alarm pheromone-based repellent was tested in field conditions. The repellent was formulated with wax for applying by pasting (Fig 5A). To test the efficacy of the formulation, the repellents were applied to hot peppers infested with *F. occidentalis* in a greenhouse (Fig 5B). Test plants without any thrips were covered with a plastic cage, which had large openings in four directions to allow the entry of nearby thrips. The repellents were then applied to the openings. In the control without the repellent, thrips arrived within a day and maintained an immigrating population. However, hot peppers treated with the repellents maintained significantly ($F$ = 192.79; df = 1, 34; $P$ < 0.0001) low levels of thrips for at least two weeks (Fig 5C).

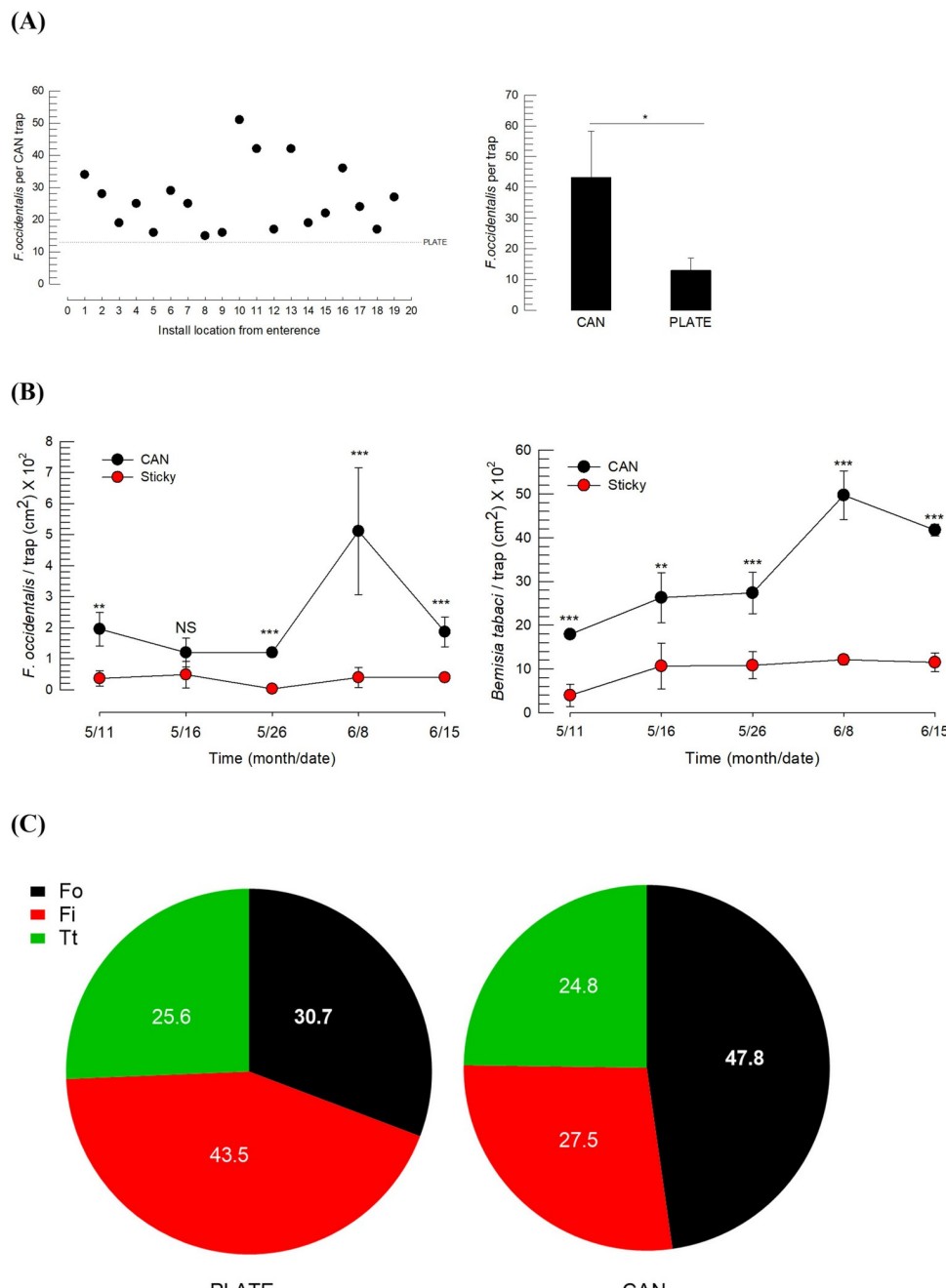

**Fig 3. Field efficacy of a yellow CAN trap containing aggregation pheromone.** (A) Enhanced trapping efficiency of the CAN trap containing aggregation pheromone compared to the yellow sticky plate (PLATE). CAN and PLATE traps used 20 and three replications, respectively. An asterisk above the standard deviation bars indicates a significant difference between two treatments at type I error = 0.05 (LSD test). (B) Enhanced pulling effect of the CAN trap on trapping *F. occidentalis* and *B. tabaci* in different seasons. Two and three asterisks above the standard deviation bars indicate significant differences between two treatments at Type I error = 0.01 and type I error = 0.0001 (LSD test), respectively. NS indicates no significant difference. (C) Composition of thrips trapped by the PLATE and CAN traps. The average species frequencies in CAN and PLATE traps were obtained from 20 and three replications, respectively.

The repellent efficacy of the wax formulation was maintained for four weeks without significant ($F$ = 1,600.00, df = 4, 10; $P$ = 0.0001) differences in the greenhouse conditions.

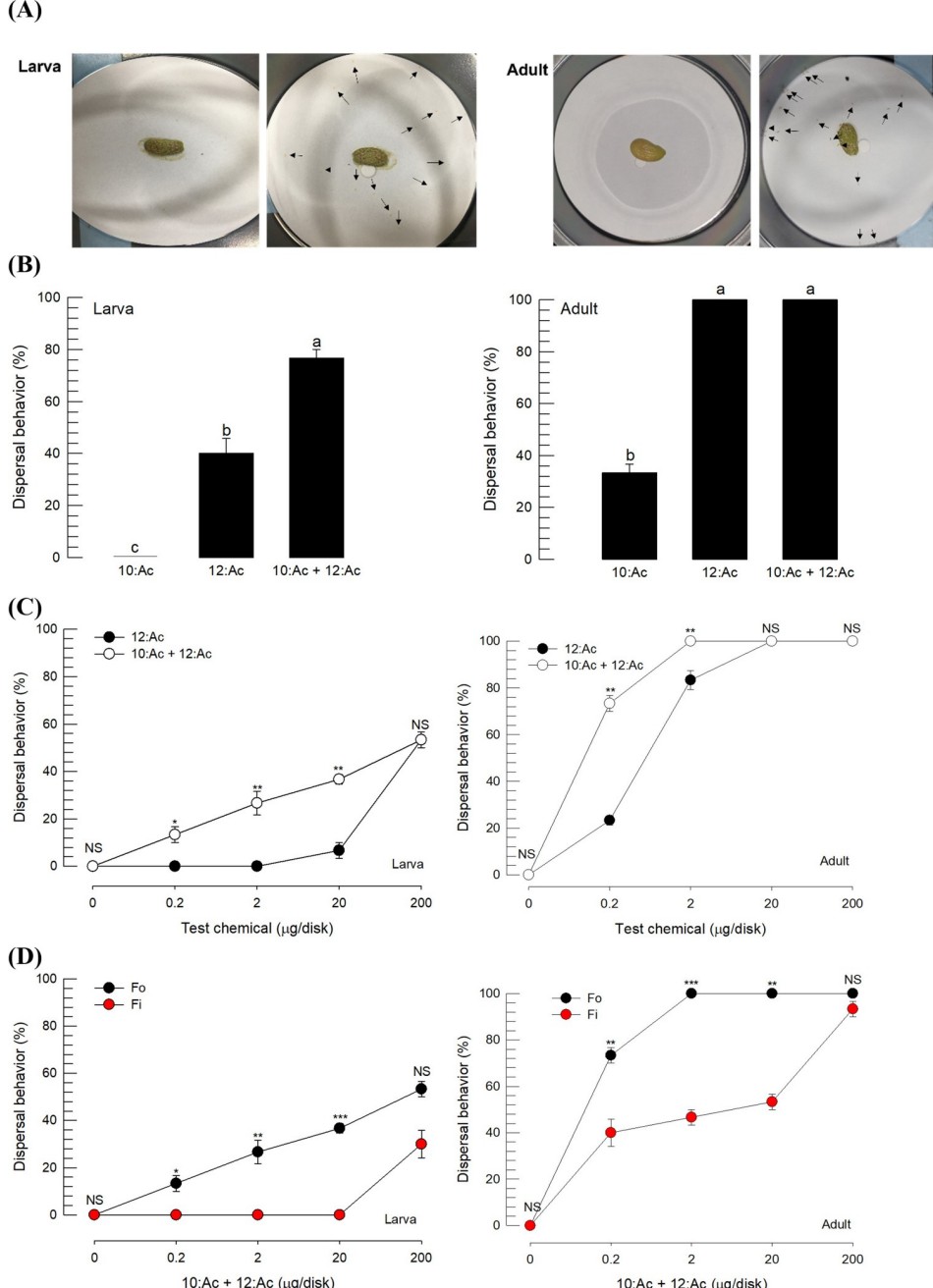

**Fig 4. Development of a thrips repellent using an alarm pheromone of *F. occidentalis*.** The alarm pheromone components were decyl acetate (10:Ac) and dodecyl acetate (12:Ac) at a 1:1.5 mass ratio. (A) Bioassay assessing repellent efficacy measured by the dispersal behavior of adults and larvae from bean diet after the test component (10 μL) was dispensed onto a filter paper disc. Arrows indicate the dispersal behavior of larvae. Different letters above the standard deviation bars indicate a significant difference between means at type I error = 0.05 (LSD test). (B) Repellent efficacy of single and dual components. (C) Dose-response of the pheromone components. (D) Comparison of dual component pheromone (alarm pheromone) efficacy in *F. occidentalis* (Fo) and *F. intonsa* (Fi). Each treatment was replicated three times. Asterisks above the standard deviation bars indicate significant difference between two treatments (LSD test): * at Type I error = 0.05, ** at Type I error = 0.01, and *** at Type I error = 0.0001. NS indicates no significant difference.

**(A)**

| Components | Weight or volume |
|---|---|
| Paraffin wax | 30 g |
| Emulsifier (Almax 3600) | 6 g |
| α-Tocopherol | 2 g |
| Jojoba oil | 2 g |
| Dodecyl acetate | 1.5 mL |
| Decyl acetate | 1 mL |
| Distilled water | 48 mL |

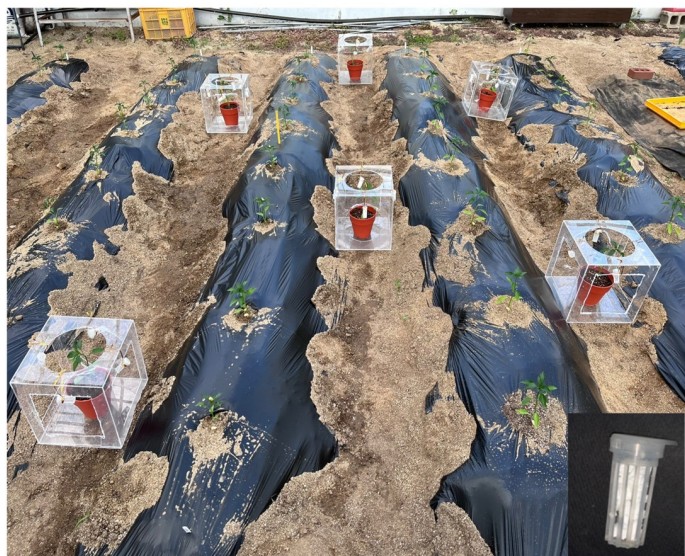

**(B)**          **(C)**

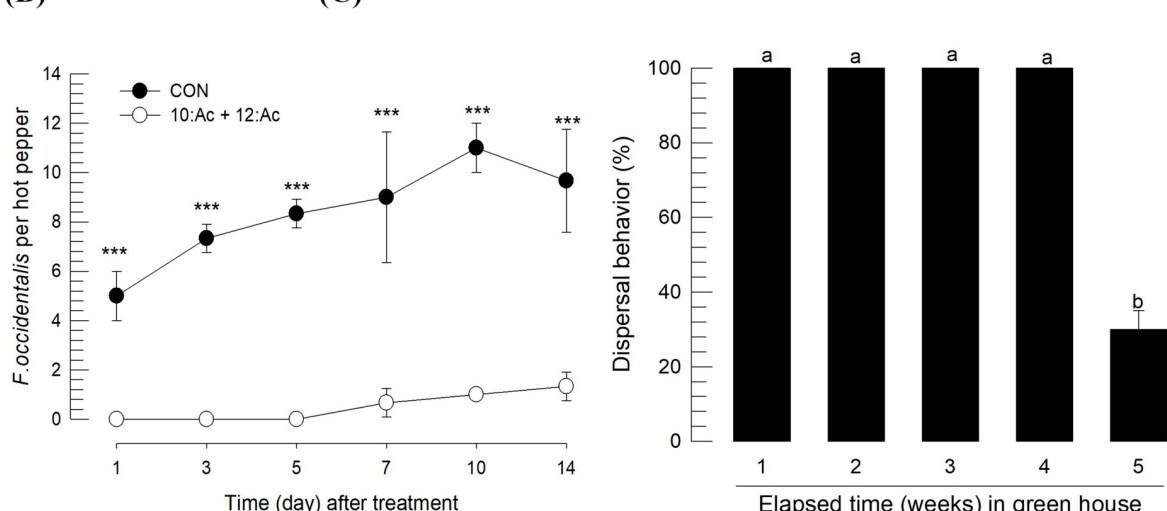

**Fig 5. Field assay of alarm pheromone-based repellent efficacy against *F. occidentalis* in a greenhouse cultivating hot peppers.** (A) Composition of the wax formulation of the repellent and installation of the wax formulations (in a small cage) in five directions around a test plant. (B) Effect of the repellent on preventing immigrating thrips for two weeks. Three asterisks above the standard deviation bars indicate a significant difference between two treatments at Type I error = 0.0001 (LSD test). (C) Determination of the effective period of the wax formulation in field conditions. Different letters above the standard deviation bars indicate a significant difference among means at Type I error = 0.05 (LSD test).

## 3.5. Application of the push-pull strategy to an agricultural farm cultivating hot peppers

The occurrence of *F. occidentalis* in three different greenhouses cultivating hot peppers in Andong was compared (Fig 6A). Push-pull control was applied to the GD greenhouse (Fig 6B), while HH and SC were negative controls and heavily treated with conventional insecticides, in which HH hot peppers were in a greenhouse, but SC peppers were in an open field.

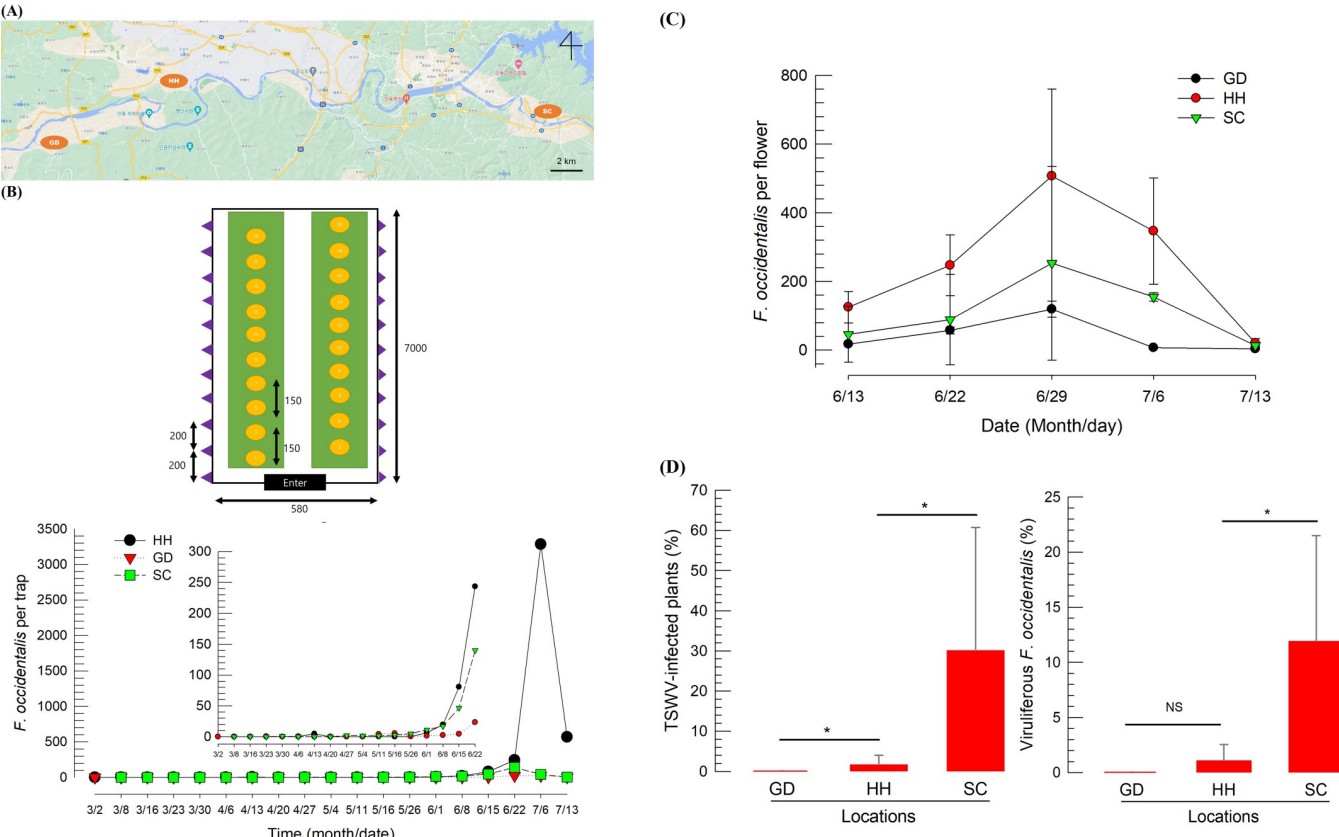

**Fig 6. Application of push-pull strategy to _F. occidentalis_ infesting hot peppers (200 plants) in a greenhouse (660 m$^2$).** (A) Deployment of repellents and CAN traps in a GD greenhouse. Adult occurrence in three sites. (B) Number of _F. occidentalis_ in flowers. Each replication represents the total number of thrips from 10 randomly selected flowers. Each measurement was replicated three times. (C) TSWV infection rates in hot peppers and _F. occidentalis_ over one month (June 13 to July 13). Each measurement represents the percentage of diseased plants out of 100 randomly chosen hosts. Each measurement was replicated three times. To determine TSWV-infected thrips, multiplex PCR analysis was performed to discriminate viruliferous thrips. In each site/collection, 30 thrips were used to analyze the viruliferous thrips for the period. An asterisk above the standard deviation bars indicates a significant difference among means at Type I error = 0.05 (LSD test). NS indicates no significant difference.

_F. occidentalis_ adults usually occur in greenhouses in late March in Andong [12]. Our current study also found that the first adults were caught in greenhouses on March 23 in HH and April 27 in GD. In contrast, the occurrence of adults was delayed and first detected in early May in an SC open field. After small peaks in April-May, the main adult peak was observed in June-July in all three field conditions. In the main peak, HH had many more adults than GD by more than 1,000-fold. The occurrence of _F. occidentalis_ adults in GD was lower than that in the SC open field. _F. occidentalis_ usually occurs at low levels because the thrips prefer greenhouse conditions over open fields [30]. To further analyze the effect of the CAN trap, thrips populations in the flowers were compared (Fig 6C). The number of _F. occidentalis_ thrips in the flowers was significantly lower in CAN trap-treated GD compared to the control farms ($F = 5.89$; df = 10, 30; $P < 0.0001$). _F. occidentalis_ is a vector for TSWV. The TSWV infection rate was significantly different between farms ($F = 8.55$; df = 2, 22; $P = 0.0049$). Indeed, no hot peppers were infected with TSWV in CAN trap-treated GD (Fig 6D). However, the control farms had TSWV-infected plants, in which the SC site showed an infection rate of more than 30%. This was further supported by the multiplex PCR analysis. No _F. occidentalis_ was infected with TSWV, whereas significant numbers of _F. occidentalis_ adults were positive for TSWV in HH and SC.

## 4. Discussion

This study proposes a push-pull control strategy using alarm and aggregation pheromones against *F. occidentalis* in greenhouse conditions. To test these novel control tactics, the push-pull effects of the chemical compounds were re-evaluated. For field assays, both pheromones were formulated as wax types for the alarm pheromone and a CAN trap for the aggregation pheromone. These formulated pheromones were applied to a practical farm and the control efficacy was compared to those of other conventional farms treated with chemical insecticides. Our laboratory and field data supported the control efficacy of push-pull control tactics against *F. occidentalis*.

To enhance the attractiveness for *F. occidentalis*, two aggregation pheromone components (LA and NMB) were re-evaluated with an analog, lavandulyl methylbutanoate (LMB). LMB was identified as an aggregation pheromone component of another flower thrips, *Megalurothrips sjostedti* [31], and onion thrips, *Thrips palmi* [32]. No significant attractiveness of LMB for either *F. occidentalis* or *F. intonsa* was seen by Y-tube olfactometry. However, they were attracted to LA and NMB. The aggregation pheromones LA and NMB are shared between *F. occidentalis* and *F. intonsa*, so species-specific blend mixtures were used to prevent cross-attraction [33]. In our assay, *F. occidentalis* preferred a 1:7 (LA:NMB) ratio to a 1:4 ratio, whereas *F. intonsa* exhibited an opposite preference. Interestingly, a significant preference was observed only in the females of both species in the Y-tube olfactometer analysis in the double-choice experiment between 1:4 and 1:7 ratios, although both sexes were attracted to both ratios in the single-choice experiments. Considering that the aggregation pheromone is basically attractive to both sexes, our assay might not detect a male preference. The lower sensitivity of males may be their preference for a specific stereoisomer because our synthetic pheromone components (S1 Fig) were mixtures of stereoisomers. The LA:NMB ratio in *F. occidentalis* was reported to be within a range of 1:0.8 ~ 1:5 measured by solid-phase microextraction (SPME) [22]. In that study, the addition of LA to NMB gave no increase relative to NMB alone, leading to the development of a single-component commercial product. In our current study, however, the inclusion of LA in the aggregation pheromone may have had some physiological significance because unlike the 1:4 and 1:7 ratios, the 1:2 mixture ratio did not significantly attract *F. occidentalis*. This suggests the role of LA in the aggregation signal of *F. occidentalis*. Two congener species of *F. occidentalis* and *F. intonsa* share flower habitats and aggregation pheromone components, in which *F. occidentalis* had 1:13 (LA:NMB) ratio while *F. intonsa* had 1:2 ratio in a Chinese population measured by SPME [34]. The significance of these different effective mixture ratios was confirmed in field experiments where a 1:8 ratio of aggregation pheromone components caught more *F. occidentalis*, whereas a 1:4 ratio attracted more *F. intonsa* [33]. Thus, the role of LA may give species-specificity to the aggregation pheromone. Furthermore, in our current assay, LA alone significantly attracted *F. occidentalis*. Thus, we determined the optimal mass trapping lure to be a mixture of LA and NMB at a 1:7 mixture ratio.

An alarm pheromone consisting of 10:Ac and 12:Ac induced significant dispersion behavior in *F. occidentalis*. It was reported that in response to the alarm pheromone, larvae exhibited escape behavior from the source by a rapid drop from host plants, whereas exposed adults exhibited take-off behavior and avoided landing in the area [35]. Indeed, the deployment of the alarm pheromone significantly prevented the invasion of the thrips to hot peppers in the current study. These facts led us to apply the alarm pheromone to the push-pull strategy to control *F. occidentalis*. Other potent repellents may be developed from other sources. For example, plant odors emitted by non-preferred hosts, such as garlic (*Allium sativum*) and celery (*Apium graveolens*), showed repelling effects on *F. occidentalis* females [36].

To apply the pheromones to field conditions, they were formulated to meet the purpose of the push-pull strategy. Alarm pheromone components were formulated using wax for fence treatment by pasting. The wax formulation modified the methods used in other lure formulations developed using moth sex pheromone and fly lure, where their efficacy was confirmed in field conditions [28, 37]. The bioactivity of the wax formulation of the alarm pheromone lasted for at least four weeks in greenhouse conditions. Aggregation pheromone was formulated by mixing with sticky material in an attract-to-kill strategy. For this purpose, a CAN trap was devised and effectively attracted thrips by its yellow color and aggregation pheromone. The question has been raised of whether thrips from nearby plants or thrips flying over plants were collected by the CAN traps. The study findings present indirect evidence that CAN traps collected thrips flying over the plants because adult thrips in the hot pepper flowers were not evenly distributed, whereas the 20 installed CAN traps collected similar quantities of thrips. This suggests that our CAN traps attracted flying thrips. A synergistic effect of the CAN trap and another biological control method using a natural enemy was also suggested because it is known that the predator *Orius laevigatus* stimulates the release of an alarm pheromone [27], which induces take-off flight in flying thrips in host plants.

Interestingly, the trapping efficiency of the yellow CAN trap for the whitefly *B. tabaci* was enhanced without an aggregation pheromone. This suggests that the aggregation pheromone of *F. occidentalis* was effective in attracting *B. tabaci*. The two aggregation pheromone components are terpenoids. Whiteflies respond to various terpenoids including sesquiterpenes (zingiberene and curcumene) and monoterperpenes (*p*-cymene, *α*-terpinene, and *α*-phellandrene), although terpenoids act as repellents [38]. It is noteworthy that LMB, one of the aggregation pheromone components in thrips, is used by a sex pheromone of the mealybug, *Phenacoccus madeirensis* [39]. Further studies assessing the sensitivity and response of *B. tabaci* to the aggregation pheromone of *F. occidentalis* need to be conducted.

The push-pull strategy was effective in suppressing the frequency of TSWV-infected hot peppers. Compared to other fields such as those in HH and SC, the thrips density was low in flowers in GD treated with the push-pull strategy, as was the number in monitoring traps. TSWV-infected hot peppers exhibit characteristic disease symptoms of ring-spots on leaves and fruit [40]. Based on this disease symptom, the survey indicated no TSWV infections in hot peppers in GD in contrast to the significant numbers of disease symptoms in the other two fields. This was further supported by the multiplex PCR results, which indicated no TSWV-infected *F. occidentalis* in GD compared to large numbers of TSWV-positive thrips in other fields. This might be explained by a local variation in viruliferous thrips as similar variations in *Thrips tabaci* carrying TSWV were seen [41]. Alternatively, the suppression of thrips occurrence by the push-pull treatment might have caused the decrease in the TSWV infection rate.

The push-pull strategy has been used for effective insect pest control [42]. A push-pull control technique against *F. occidentalis* was proposed using UV-reflective mulch as a pushing factor to disrupt thrips colonization on host plants along with a companion plant as a pulling factor to attract the predators and trap the thrips [43]. Alternatively, the use of semiochemicals such as attractants and repellents has been considered for developing a push-pull strategy [44]. Kirk et al. [23] proposed *F. occidentalis* control using pheromones and other semiochemicals. This study demonstrated novel push-pull tactics using aggregation/alarm pheromones (Fig 7) to control *F. occidentalis* in greenhouse conditions. This control technique effectively suppressed the outbreak of *F. occidentalis* in a field assay and also prevented the occurrence of plant disease caused by TSWV in hot peppers. Although our current study directly evaluated the pulling efficacy of the attracting CAN trap, the pushing efficacy of the repellent fence treatment was indirectly predicted from a small-scale greenhouse experiment. Subsequent studies

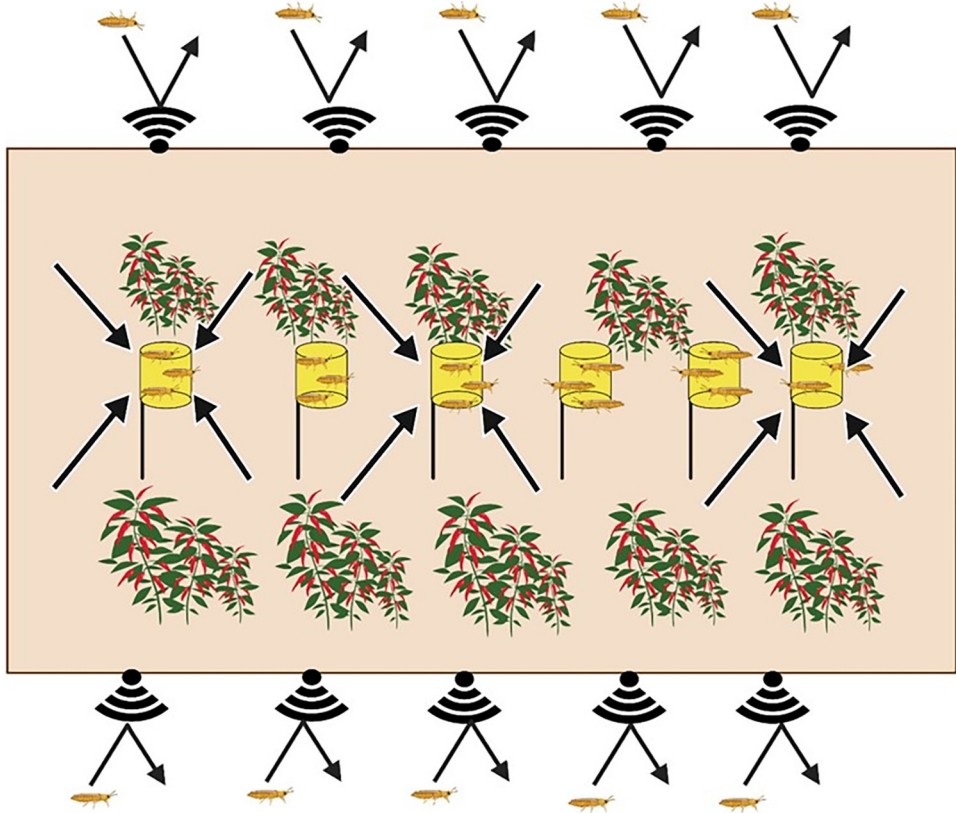

**Fig 7. A working model of the push-pull strategy against *F. occidentalis* using repellent (sound signal logo) and CAN trap (yellow can) in a greenhouse cultivating hot peppers.** Arrows indicate the push and pull moving directions.

are needed to assess thrips occurrence outside of a test greenhouse treated with the alarm pheromone to improve the estimation of the pushing efficacy in field conditions.

## Supporting information

**S1 Fig. Chemical synthesis of the aggregation pheromone components of thrips: Lavandulyl acetate (LA, 2), lavandulyl methylbutanoate (LMB, 3), and neryl methylbutanoate (NMB, 5).** LA and LMB were synthesized from the precursor, lavandulol (1) in dichloromethane, which was reacted with acetyl chloride to yield LA, and was reacted with isovaleryl chloride to yield LMB, respectively, under the catalytic activity of trimethylamine. NMB was reacted with geraniol ('4') and isovaleryl chloride under a catalytic activity of trimethylamine. The purity of the compounds was analyzed by gas chromatography (8860 GC, Agilent, Santa Clara, CA, USA) with a DB-1 column (15 m × 0.350 mm, Agilent) at an oven temperature of 280˚C and a flow rate of 1.0 mL/min.
(DOCX)

**S2 Fig. Tests of two flower thrips (*F. occidentalis* ('Fo') and *F. intonsa* ('Fi')) to aggregation pheromone components using a Y-tube olfactometer.** The test pheromone components included LA (lavandulyl acetate), LMB (lavandulyl methylbutanoate), and NMB (neryl methylbutanoate). Each response (experimental unit) used 10 adults (< 3-days-old after emergence).

Each treatment was replicated four times.
(DOCX)

## Acknowledgments

We thank Jean Lee and Juan Hong in our laboratory to provide thrips for our analysis.

## Author Contributions

**Conceptualization:** Yonggyun Kim.

**Data curation:** Chul-Young Kim.

**Formal analysis:** Chul-Young Kim.

**Funding acquisition:** Yonggyun Kim.

**Investigation:** Chul-Young Kim, Falguni Khan, Yonggyun Kim.

**Methodology:** Chul-Young Kim, Falguni Khan.

**Project administration:** Yonggyun Kim.

**Resources:** Chul-Young Kim, Falguni Khan.

**Software:** Chul-Young Kim, Falguni Khan.

**Supervision:** Yonggyun Kim.

**Validation:** Chul-Young Kim.

**Visualization:** Chul-Young Kim.

**Writing – original draft:** Chul-Young Kim, Yonggyun Kim.

**Writing – review & editing:** Yonggyun Kim.

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
