## [Decision Letter · Decision Letter 0]

29 Sep 2022

PONE-D-22-23616A push-pull strategy to control the western flower thrips, Frankliniella occidentalis, using alarm and aggregation pheromonesPLOS ONE

Dear Dr. Kim,

Thank you for submitting your manuscript to PLOS ONE. After careful consideration, we feel that it has merit but does not fully meet PLOS ONE’s publication criteria as it currently stands. Therefore, we invite you to submit a revised version of the manuscript that addresses the points raised during the review process.

We look forward to receiving your revised manuscript.

Kind regards,

Ramzi Mansour

Academic Editor

PLOS ONE

Journal Requirements:

"This work was conducted with the support of the Cooperative Research Program for Agriculture Science & Technology Development (Project No. PJ01578901) funded by the Rural Development Administration, Republic of Korea."

5. Please amend your list of authors on the manuscript to ensure that each author is linked to an affiliation. Authors’ affiliations should reflect the institution where the work was done (if authors moved subsequently, you can also list the new affiliation stating “current affiliation:….” as necessary).

6. We note that Figure 6A in your submission contain map images which may be copyrighted. All PLOS content is published under the Creative Commons Attribution License (CC BY 4.0), which means that the manuscript, images, and Supporting Information files will be freely available online, and any third party is permitted to access, download, copy, distribute, and use these materials in any way, even commercially, with proper attribution. For these reasons, we cannot publish previously copyrighted maps or satellite images created using proprietary data, such as Google software (Google Maps, Street View, and Earth). For more information, see our copyright guidelines: http://journals.plos.org/plosone/s/licenses-and-copyright.

a. You may seek permission from the original copyright holder of Figure 6A to publish the content specifically under the CC BY 4.0 license.  

Reviewers' comments:

Reviewer's Responses to Questions

**Comments to the Author**

1. Is the manuscript technically sound, and do the data support the conclusions?

Reviewer #1: Yes

Reviewer #2: Yes

Reviewer #3: Partly

2. Has the statistical analysis been performed appropriately and rigorously? 

Reviewer #1: Yes

Reviewer #2: Yes

Reviewer #3: No

3. Have the authors made all data underlying the findings in their manuscript fully available?

Reviewer #1: Yes

Reviewer #2: Yes

Reviewer #3: No

4. Is the manuscript presented in an intelligible fashion and written in standard English?

Reviewer #1: Yes

Reviewer #2: Yes

Reviewer #3: Yes

5. Review Comments to the Author

Reviewer #1: I read the manuscript by Yonggyun Kim et al. with interest. The paper aims to develop a non-chemical technique to control F. occidentalis infesting hot peppers cultivated in greenhouses in Korea. The method was based on behavioral control using an alarm pheromone to prevent the entry of the thrips into greenhouses and an aggregation pheromone for mass trapping inside the greenhouses. This technique effectively suppressed thrips populations in field conditions and also prevented the occurrence of plant disease caused by TSWV in hot peppers. In my opinion, I think is a nice piece of work and original, however there are some improvements to be made especially in materials and Methods section.

Introduction

I believe that you need a much strong justification for studying the western flower thrips, Frankliniella occidentalis as pest I did not find any strong justification for using F. occidentalis compared to other insects such as aphids for example!

Materials and methods

The overall level of details for the Materials and Methods is appropriate, however there are some informations should be add in the text:

Line 98 page 5: Please precise if it was the same laboratory conditions described below.

Line 103-105 page 5: please give the purity of the molecules: Lavandulyl acetate (LA), lavandulyl methylbutanoate (LMB), and neryl methylbutanoate (NMB).

Line 108-110 page 6: Purity of molecules are missing here.

Line 114 page 6: add the purity of the alarm pheromone components

Line 115: please correct Signal-Aldrich >> Sigma-Aldrich

Line 125: I did not find any justification for using hexane as solvent please explain.

Line 132: Give the purity of α-tocopherol.

Line 138: explain the use of a dark room

Line 143: here, normally the Y-tube should be cleaned with methanol after each replication not only before and after the bio assay.

Line 143: give the sex of individuals: only females? Please justify

Line 182: information about pesticides used in fields are missing here. Please add the active molecules of these insecticides.

Line 206-211: add references about used primers.

Line 218: reference is missing.

Line 220: Do you check the normality of values before using the one-way analysis of variance (ANOVA) ? please add the information.

Reviewer #2: The manuscript does not provide a proper paragraph of Conclusions which must be drawn appropriately based on the data presented in the text. The data obtained in the study support the conclusions. All the experiments, trials have been conducted rigorously , with appropriate control and replication.So that also the descriptions of Material and Methods must be precse and complete. Although the manuscript has been presented in a correct and standard English, it might be useful to ask for a revision of the text by a mother language lecturer.

Reviewer #3: The article in review is titled “A push-pull strategy to control the western flower thrips, Frankliniella occidentalis, using alarm and aggregation pheromones”. This article presents a novel push-pull system for the management of Frankliniella occidentalis thrips using an interesting and new combination of pheromones and traps. The experimental design was simple and straightforward, and acceptable. The article is mostly well-written and has a strong introduction supporting the importance of the experiment. There are many components to this research involving different scientific techniques and methods, which are interesting but may have contributed to a few issues which need to be addressed before the publication process can progress.

There are some areas in the methods section which need clarification, and it seems as though the results section contains much of the information which fits better into the methods section. There are major problems with the reporting of the statistical results and the description of the statistical analyses used. The statistical values reported do not match the tests that were reported to be used in the methods section. Additionally, the discussion section could use some expanding to include discussion of the study’s limitations, future directions, and more summarizing of the results highlighting the importance for the field of study. More details on suggested areas for improvement can be found below.

Overall, I believe that this is an interesting and important study for the field of integrated pest management, but it requires significant revisions before it can be considered for publication. Specifically, there are some major issues involving the choice and reporting of statistical analyses that require extra attention, and the discussion section is weak.

Areas for improvement

Major

1.Results section appears to be missing the F-values, and degrees of freedom in the reporting of the results of the ANOVA. There appear to be sections of the results section that might actually belong in the methods section, for example the details about the trap types and configuration.

2.The results reported for the trap comparison test use a Chi-squared value rather than an F-value which would be expected from the ANOVA that was purported to be used to analyze the results. The statistics reported in the results section do not match the statistical tests that were delineated in the methods section. Lines 304-306 only a P-value is reported, what statistical test was used here and what was the associated value? Additionally, were the data tested to confirm that they meet the assumptions of the statistical tests chosen for analysis? Did the data transformation correct for normality?

3. In the discussion section there is a lack of discussion about any potential shortcomings/weaknesses of the study and the discussion section should have more time spent discussing future directions.

Minor

1.The abstract could use more information regarding the results of the study.

2.In the methods section under thrips rearing, it is stated that the thrips were fed a kidney bean diet for five days after germination. This could use some clarification as I believe the authors meant after hatching.

3.Y-tube methods (line 137-146) could use more explanation. It says that thrips were used three days after emergence, is that referring to emergence from the egg (hatching), or emergence as adults from the pupal stage? Additionally, the methods state that each test used 10-30 individuals, were these individuals tested separately (one at a time in the olfactometer) or were they all places into the olfactometer at the same time, please clarify.

4.Line 189-190 discussing the sampling of flowers states: “in flowers in 10 randomly selected flowers was counted in three fields three times a week for 190 one month (June 13 � July 13) once in a week”. This statement is confusing, was the sampling conducted three times a week or one time per week?

5.Line 222- should be arcsine perhaps?

6.In discussing the results of the field study of the push and pull components, it is stated that the Frankliniella occidentalis numbers are lower in the experimental greenhouse, but what were the results for each life stage and sex? Was it only the total number of all life stages and males and females that were reduced, or was there a difference in the response of the different life stages (larvae versus adults) or sexes?

6. PLOS authors have the option to publish the peer review history of their article (what does this mean?). If published, this will include your full peer review and any attached files.

Reviewer #1: No

Reviewer #2: No

Reviewer #3: No

---

## [Author Response · Author response to Decision Letter 0]

11 Oct 2022

[Reviewer #1]

Comment #1-1: Introduction. I believe that you need a much strong justification for studying the western flower thrips, Frankliniella occidentalis as pest I did not find any strong justification for using F. occidentalis compared to other insects such as aphids for example!

Response: The thrips as a serious pest is explained by direct feeding damage and indirect TSWV transmission. We did not find the direct analysis on the statistical data of the relative damage compared to aphids. Thus we did not add the further information.

Comment #1-2: Materials and methods. The overall level of details for the Materials and Methods is appropriate, however there are some informations should be add in the text:

Line 98 page 5: Please precise if it was the same laboratory conditions described below.

Response: It is clarified as follows: “…. reared in the laboratory conditions.” 

Comment #1-3: Line 103-105 page 5: please give the purity of the molecules: Lavandulyl acetate (LA), lavandulyl methylbutanoate (LMB), and neryl methylbutanoate (NMB).

Response: Added

Comment #1-4: Line 108-110 page 6: Purity of molecules are missing here.

Response: Added

Comment #1-5: Line 114 page 6: add the purity of the alarm pheromone components

Response: added

Comment #1-6: Line 115: please correct Signal-Aldrich >> Sigma-Aldrich

Response: Corrected as suggested

Comment #1-7: Line 125: I did not find any justification for using hexane as solvent please explain.

Response: It is rephrased as follows: “To mix the pheromone with sticky material, 2 mg of pheromone components and 12 g of sticky material (Tanglefoot, Contech Electronics, Rochester, NY, USA) were dissolved in 6 mL of hexane. This mixture was then poured to a sticky plastic zipper bag (30 � 30 cm, HG Pack, Gunpo, Korea).”

Comment #1-8: Line 132: Give the purity of α-tocopherol.

Response: Added

Comment #1-9: Line 138: explain the use of a dark room

Response: Added as follows: “to avoid visual cue for the choice test”

Comment #1-10: Line 143: here, normally the Y-tube should be cleaned with methanol after each replication not only before and after the bio assay.

Response: As you see, we wrote the cleaning steps.

Comment #1-11: Line 143: give the sex of individuals: only females? Please justify

Response: Sex was discriminated. This information is added as follows: “Each test used 10 � 30 individuals of each sex and was replicated four times by changing the source and control for replication.”

Comment #1-12: Line 182: information about pesticides used in fields are missing here. Please add the active molecules of these insecticides.

Response: Added as follows: “Both fields were frequently treated with chemical insecticides including spinosad during the assay.”

Comment #1-13: Line 206-211: add references about used primers.

Response: Added

Comment #1-14: Line 218: reference is missing.

Response: Added

Comment #1-15: Line 220: Do you check the normality of values before using the one-way analysis of variance (ANOVA) ? please add the information.

Response: Added as follows “Percentage data were arsine-transformed and confirmed in normality using PROC UNIVARIATE in the SAS program [29].”

[Reviewer #2]

Comment #2-1: The manuscript does not provide a proper paragraph of Conclusions which must be drawn appropriately based on the data presented in the text. The data obtained in the study support the conclusions. All the experiments, trials have been conducted rigorously , with appropriate control and replication. So that also the descriptions of Material and Methods must be precse and complete. Although the manuscript has been presented in a correct and standard English, it might be useful to ask for a revision of the text by a mother language lecturer.

Response: 

(1) At the end of Discussion, the conclusion is described as follows: “This control technique effectively suppressed the outbreak of F. occidentalis in a field assay and also prevented the occurrence of plant disease caused by TSWV in hot peppers. Although our current study directly evaluated the pulling efficacy of the attracting CAN trap, the pushing efficacy of the repellent fence treatment was indirectly predicted from a small-scale greenhouse experiment. Subsequent studies are needed to assess thrips occurrence outside of a test greenhouse treated with the alarm pheromone to improve the estimation of the pushing efficacy in field conditions.” This also explains a limitation of this technology in field conditions.

(2) Some details are added in M&M. These are marked in red color. 

(3) Text is cleaned by a English-editing company, Harrisco, co.

 

[Reviewer #3]

Comment #3-1: Results section appears to be missing the F-values, and degrees of freedom in the reporting of the results of the ANOVA. There appear to be sections of the results section that might actually belong in the methods section, for example the details about the trap types and configuration.

Response: All F values are added. The trap was newly designed. Thus in M&M, the structure and detailed configuration were explained. In Results, the background rationale to construct the trap was explained. 

Comment #3-2: The results reported for the trap comparison test use a Chi-squared value rather than an F-value which would be expected from the ANOVA that was purported to be used to analyze the results. The statistics reported in the results section do not match the statistical tests that were delineated in the methods section. Lines 304-306 only a P-value is reported, what statistical test was used here and what was the associated value? Additionally, were the data tested to confirm that they meet the assumptions of the statistical tests chosen for analysis? Did the data transformation correct for normality?

Response: All the percent data were transformed by arcsine and tested by PROC UNIVARIATE for normality. This information is added. All the F test values are added.

Comment #3-3: In the discussion section there is a lack of discussion about any potential shortcomings/weaknesses of the study and the discussion section should have more time spent discussing future directions.

Response: At the end of Discussion, the information is described as follows: “This control technique effectively suppressed the outbreak of F. occidentalis in a field assay and also prevented the occurrence of plant disease caused by TSWV in hot peppers. Although our current study directly evaluated the pulling efficacy of the attracting CAN trap, the pushing efficacy of the repellent fence treatment was indirectly predicted from a small-scale greenhouse experiment. Subsequent studies are needed to assess thrips occurrence outside of a test greenhouse treated with the alarm pheromone to improve the estimation of the pushing efficacy in field conditions.” This also explains a limitation of this technology in field conditions.

Comment #3-4: The abstract could use more information regarding the results of the study.

Response: The additional information is added as follows: “Field assay demonstrated the efficacy of the push-pull tactics by reducing thrips density in flowers of the hot peppers as well as in the monitoring traps. Especially, the enhanced mass trapping to the CAN trap compared to the conventional yellow sticky trap led to significant reduction in the thrips population. This novel push-pull technique would be applicable to effectively control F. occidentalis in field conditions.”

Comment #3-5: In the methods section under thrips rearing, it is stated that the thrips were fed a kidney bean diet for five days after germination. This could use some clarification as I believe the authors meant after hatching.

Response: Corrected as follows: “All thrips were fed a kidney bean (Phaseolus coccineus L.) diet according to the method described in our earlier study [12].”

Comment #3-6: Y-tube methods (line 137-146) could use more explanation. It says that thrips were used three days after emergence, is that referring to emergence from the egg (hatching), or emergence as adults from the pupal stage? Additionally, the methods state that each test used 10-30 individuals, were these individuals tested separately (one at a time in the olfactometer) or were they all places into the olfactometer at the same time, please clarify.

Response: Some uncertainty is rephrased as follows: “A Y-type olfactometer (the main Y-tube length, 5 cm; two branches 2 cm long; a 45o angle between the branches; inner branch diameter, 5 cm) was placed in a dark room at 25 ± 1oC to avoid visual cue for the choice test. A flow of clean (charcoal-filtered) air at a rate of 0.6 L/min was split and passed through two glass vessels containing either an odor source or control (hexane) stimuli and entered each of the branches of the Y-tube. An air supply system (Power Air Pump, Seoul, Korea) was used for air filtration and flow rate control. Before and after the bioassay, the Y-tube was cleaned with 100% methanol. Test thrips were used three days after adult emergence. Each test used 10 � 30 individuals of each sex and was replicated four times by changing the source and control for replication.”

Comment #3-7: Line 189-190 discussing the sampling of flowers states: “in flowers in 10 randomly selected flowers was counted in three fields three times a week for 190 one month (June 13 � July 13) once in a week”. This statement is confusing, was the sampling conducted three times a week or one time per week?

Response: This is rephrased as follows: “Number of F. occidentalis in flowers was counted from 10 randomly selected flowers with three replications in each field. The monitoring was performed once a week for one month (June 13 � July 13).”

Comment #3-8: Line 222- should be arcsine perhaps?

Response: Corrected as suggested

Comment #3-9: In discussing the results of the field study of the push and pull components, it is stated that the Frankliniella occidentalis numbers are lower in the experimental greenhouse, but what were the results for each life stage and sex? Was it only the total number of all life stages and males and females that were reduced, or was there a difference in the response of the different life stages (larvae versus adults) or sexes?

Response: We did not examine larval density because we assessed the efficacy mostly based on the trapping adults on the yellow sticky trap. However, the figure 6C suggests the total number of larvae and adults. Thus the reduction by the push-pull can be explained by the total reduction not only by adults.

---

## [Decision Letter · Decision Letter 1]

8 Dec 2022

PONE-D-22-23616R1A push-pull strategy to control the western flower thrips, Frankliniella occidentalis, using alarm and aggregation pheromonesPLOS ONE

Dear Dr. Kim,

Thank you for submitting your manuscript to PLOS ONE. After careful consideration, we feel that it has merit but does not fully meet PLOS ONE’s publication criteria as it currently stands. Therefore, we invite you to submit a revised version of the manuscript that addresses the points raised during the review process.

We look forward to receiving your revised manuscript.

Kind regards,

Ramzi Mansour

Academic Editor

PLOS ONE

Journal Requirements:

Reviewers' comments:

Reviewer's Responses to Questions

**Comments to the Author**

1. If the authors have adequately addressed your comments raised in a previous round of review and you feel that this manuscript is now acceptable for publication, you may indicate that here to bypass the “Comments to the Author” section, enter your conflict of interest statement in the “Confidential to Editor” section, and submit your "Accept" recommendation.

Reviewer #1: All comments have been addressed

Reviewer #2: All comments have been addressed

Reviewer #3: (No Response)

2. Is the manuscript technically sound, and do the data support the conclusions?

Reviewer #1: Yes

Reviewer #2: Yes

Reviewer #3: Yes

3. Has the statistical analysis been performed appropriately and rigorously? 

Reviewer #1: Yes

Reviewer #2: Yes

Reviewer #3: I Don't Know

4. Have the authors made all data underlying the findings in their manuscript fully available?

Reviewer #1: Yes

Reviewer #2: Yes

Reviewer #3: No

5. Is the manuscript presented in an intelligible fashion and written in standard English?

Reviewer #1: Yes

Reviewer #2: Yes

Reviewer #3: Yes

6. Review Comments to the Author

Reviewer #1: all corrections were done. I feel that this manuscript A push-pull strategy to control the western flower thrips, Frankliniella occidentalis,using alarm and aggregation pheromones is now acceptable for publication

Reviewer #2: Dear Authors,

I read carefully the revised text of your mns PONE -D-22-23616, titled " A push-pull strategy to control the Western Flower Thrips, Frankliniella occidentalis,using alarm and aggregation pheromones". Even I read the replies you provided to each reviewers. I appreciated the effort done in order to justify and preserve your previous "draft", or to accept and change some parts as suggested by reviewers. In particular, I am satisfied for the adding of several new data to minor/major issues recorded in M& M and Results Sections. Such as detailed records regarding some texts , i.e. in subprgrs. 2.4, 2.5, 2.9 that more improved the data presentation , technical clarification , i.e. 2.3, and statistical analysis , i.e. 2.11. Details were added also to Results Section , i.e. F and P values in 3.1, 3.2 and 3.4 subparagraphs.

The new version of Discussion Section has more clearly recorded the main steps obtained in the technical part of this study , such as the control efficacy of push-pull tactics against Frankliniella occidentalis . Also the actractiveness of the aggregation pheromone compounds has been critically outlined for the other thrips species considered. Finally, the future perspectives of the novel push-pull tactics in the control of pest thrips might be proposed for open field applications.The well presented comments let me satisfied of the new version of the text, even without a separated Conclusions Section. Here, I attached the pdf of the revised version of the text that I consider suitable for publication.

Reviewer #3: Section 3-3 in the results section still reports statistical values (t-test and chi-squared) that are not mentioned in the methods section for statistical analyses performed. Are these the post-hoc analyses, or were they just forgotten in the methods section?

At the end of section 3-3 there is an F-value reported of 1600.00. This is an exceptionally high F-value, is that the accurate F-value or is the decimal point in the wrong place, for example should it actually be 160 or 16? If it is indeed 1600, that is very impressive!

All other comments have been addressed, the manuscript looks great.

7. PLOS authors have the option to publish the peer review history of their article (what does this mean?). If published, this will include your full peer review and any attached files.

Reviewer #1: No

Reviewer #2: No

Reviewer #3: **Yes: **Kara Tyler-Julian

---

## [Author Response · Author response to Decision Letter 1]

8 Dec 2022

[Reviewer #1]

Comment: All corrections were done. I feel that this manuscript A push-pull strategy to control the western flower thrips, Frankliniella occidentalis,using alarm and aggregation pheromones is now acceptable for publication

Response: Thank you for a nice review!

[Reviewer #2]

Comment: I read carefully the revised text of your mns PONE -D-22-23616, titled " A push-pull strategy to control the Western Flower Thrips, Frankliniella occidentalis,using alarm and aggregation pheromones". Even I read the replies you provided to each reviewers. I appreciated the effort done in order to justify and preserve your previous "draft", or to accept and change some parts as suggested by reviewers. In particular, I am satisfied for the adding of several new data to minor/major issues recorded in M& M and Results Sections. Such as detailed records regarding some texts , i.e. in subprgrs. 2.4, 2.5, 2.9 that more improved the data presentation , technical clarification , i.e. 2.3, and statistical analysis , i.e. 2.11. Details were added also to Results Section , i.e. F and P values in 3.1, 3.2 and 3.4 subparagraphs. The new version of Discussion Section has more clearly recorded the main steps obtained in the technical part of this study, such as the control efficacy of push-pull tactics against Frankliniella occidentalis. Also the actractiveness of the aggregation pheromone compounds has been critically outlined for the other thrips species considered. Finally, the future perspectives of the novel push-pull tactics in the control of pest thrips might be proposed for open field applications. The well presented comments let me satisfied of the new version of the text, even without a separated Conclusions Section. Here, I attached the pdf of the revised version of the text that I consider suitable for publication.

Response: Thank you for a nice review!

[Reviewer #3]

Comment: Section 3-3 in the results section still reports statistical values (t-test and chi-squared) that are not mentioned in the methods section for statistical analyses performed. Are these the post-hoc analyses, or were they just forgotten in the methods section?

Response: The stat method is now added to the M&M.

Comment: At the end of section 3-3 there is an F-value reported of 1600.00. This is an exceptionally high F-value, is that the accurate F-value or is the decimal point in the wrong place, for example should it actually be 160 or 16? If it is indeed 1600, that is very impressive!

Response: We calculated the F value again and confirmed.

Comment: All other comments have been addressed, the manuscript looks great.

Response: Thank you for a nice review!

---

## [Editor Report · Decision Letter 2]

12 Dec 2022

A push-pull strategy to control the western flower thrips, Frankliniella occidentalis, using alarm and aggregation pheromones

PONE-D-22-23616R2

Dear Dr. Kim,

We’re pleased to inform you that your manuscript has been judged scientifically suitable for publication and will be formally accepted for publication once it meets all outstanding technical requirements.

Kind regards,

Ramzi Mansour

Academic Editor

PLOS ONE

---

## [Editor Report · Acceptance letter]

16 Dec 2022

PONE-D-22-23616R2 

A push-pull strategy to control the western flower thrips, *Frankliniella occidentalis*, using alarm and aggregation pheromones 

Dear Dr. Kim:

I'm pleased to inform you that your manuscript has been deemed suitable for publication in PLOS ONE. Congratulations! Your manuscript is now with our production department. 

Kind regards, 

on behalf of

Dr. Ramzi Mansour 

Academic Editor

PLOS ONE